

# Magnification in preclinical procedures: effect on muscle activity and angular deviations of the neck and trunk

Júlia Margato Pazos[1], Ana Flávia Ribeiro Monteiro Fernandes[1], Edson Donizetti Verri[2], Guilherme Gallo Costa Gomes[2], Simone Cecílio Hallak Regalo[2] and Patricia Petromilli Nordi Sasso Garcia[1]

[1] Department of Social Dentistry, School of Dentistry of Araraquara, Universidade Estadual Paulista, Araraquara, São Paulo, Brazil
[2] Department of Basic and Oral Biology, School of Dentistry of Ribeirão Preto, Universidade de São Paulo, Ribeirão Preto, São Paulo, Brazil

Corresponding author
Patricia Petromilli Nordi Sasso Garcia, patricia.garcia@unesp.br

## ABSTRACT

**Objectives.** This study aimed to assess the effects of different magnification systems on the angular deviations of the neck and trunk and the muscle activities of the upper back and neck during preclinical cavity preparation.

**Methods.** This was an experimental laboratory study, with the angular deviations from the neutral positions of the neck and trunk and the activities of the bilateral upper back (the descending and ascending trapezius) and neck (sternocleidomastoid) muscles as the dependent variables. The independent variables were the different magnification systems used (Simple, Galilean, and Keplerian loupes, with direct vision as the control) and prepared teeth (teeth 16, 26, 36, and 46). A dental mannequin phantom head with artificial resin teeth was used, and Class I cavity preparations for composite resin were performed on teeth 16, 26, 36, and 46 using a 1012 round diamond bur at low speed. To analyze the angular deviations, the postures adopted during the procedure were recorded using a tripod-mounted camera positioned to provide a lateral view of the operator. A trained researcher measured the angular deviations using the software entitled "Software for Postural Assessment"—SAPO (version 0.69). Bilateral muscle activity was assessed using surface electromyography. Descriptive statistical analysis was performed, and after verifying the assumptions of normality and homoscedasticity, two-way analysis of variance and the Tukey and Games-Howell post-hoc tests were used to compare the data ($\alpha$=0.05).

**Results.** The angular deviation from the neutral position of the neck was found to be significantly higher during cavity preparations performed with the naked eye and the Simple loupe, irrespective of the prepared tooth. With regard to tooth location, the angular deviation of the neck was significantly greater during cavity preparation on teeth 16 and 26, and the angular deviation of the trunk was significantly greater during cavity preparation on tooth 26, regardless of the magnification system used. There were significant differences in right sternocleidomastoid muscle activity between the Simple, Galilean, and Keplerian loupes, with activity being the lowest for the Galilean loupe ($p = 0.008$). There were no significant differences in left sternocleidomastoid muscle activity between the loupes, regardless of the prepared tooth ($p = 0.077$). The activities of the bilateral descending trapezius and the right ascending trapezius muscles were significantly lower when the Galilean loupe was used ($p < 0.010$).
**Conclusion**. These results suggest that the Galilean loupe resulted in lower muscle activity in the neck and back regions and that the Galilean and Keplerian loupes resulted in less angular deviations of the neck and trunk during cavity preparation.

# INTRODUCTION

Musculoskeletal disorders are a significant problem affecting dental professionals and students (*Garcia et al., 2012*; *Corrocher et al., 2014*; *Garcia et al., 2015*; *Lietz, Ulusoy & Nienhaus, 2020*; *Braga et al., 2021*), with repetitive movements, static muscle work, maintaining poor posture for prolonged periods of time, and exposure to pressure and vibration, among others, being the main risk factors (*Onety et al., 2014*; *Hayes et al., 2016*; *Lietz, Ulusoy & Nienhaus, 2020*; *Bud, Pop & Cîmpean, 2023*). In fact, the maintenance of inappropriate posture is an occupational risk in dentistry from the professional training period onwards (*Hoerler et al., 2012*). Difficulties in viewing and accessing the operative field (*Chang, 2002*; *Garcia et al., 2017*) make dental professionals bend to be closer to the area being treated, resulting in repeated inclination and rotation of the head, neck, and trunk, especially towards the dominant working side (*Valachi & Valachi, 2003*).

Inappropriate spinal postures involving large angular deviations from the neutral position cause fatigue in the muscle groups of the neck, shoulders, and trunk (*Valachi, 2009*; *Lietz, Ulusoy & Nienhaus, 2020*). Consequently, other muscles are also overloaded due to the need to stabilize the affected regions and thus have to perform functions for which they were not designed. This overload can cause ischemia, thinning, and pain, all of which are factors related to the development of musculoskeletal disorders.

To minimize this muscle overload associated with the constant movements of dental professionals to be closer to the operative field, several strategies can be adopted, such as the adoption of more ergonomic work habits, use of ergonomic dental stools, which can help with the maintenance of a neutral position with lumbar lordosis (*De Bruyne et al., 2016*; *Plessas & Bernardes Delgado, 2018*; *Lietz, Ulusoy & Nienhaus, 2020*) proper adjustment of the dental stool and patient chair to reduce dentist's neck angulation (*Lietz, Ulusoy & Nienhaus, 2020*), and the use of devices that improve visualization (*La Delfa et al., 2016*; *Maggio, Villegas & Blatz, 2011*; *Carpentier et al., 2019*), such as operating microscope (*Brown, Qualtrough & McLean, 2020*) and magnification loupes (*Lietz, Ulusoy & Nienhaus, 2020*; *Braga et al., 2021*). Among these, magnification loupes stand out because they allow the operator to maintain an appropriate distance between their eyes and the patient's mouth without excessive trunk inclination and forward flexion of the head (*Farook et al., 2013*; *Congdon, Tolle & Darby, 2012*; *Carpentier et al., 2019*; *Lietz, Ulusoy & Nienhaus, 2020*; *Braga et al., 2021*; *Wajngarten et al., 2021*; *Pazos et al., 2022*). Several studies based on self-administered questionnaires (*Farook et al., 2013*)  as well as observational methods (*Carpentier et al., 2019*; *Pazos et al., 2020*) have suggested that the use of magnification

in dentistry leads to improved working posture (*Eichenberger et al., 2011*; *Carpentier et al., 2019*; *Pazos et al., 2020*; *Pazos et al., 2022*). Nevertheless, although these methods have provided important results, direct methods, such as electromyography (EMG), allow a more detailed and precise assessment of the activities of the muscle groups involved in the working postures adopted during clinical procedures (*Onety et al., 2014*; *Pazos et al., 2022*). Thus, their use would allow a better assessment of magnification devices in terms of the minimization of stress on the neck and back muscles.

In fact, a combination of observational and direct methods would be optimal for postural evaluation as well as facilitating the development of magnification systems that can improve the working posture of dentists, which would greatly contribute to the field of occupational health in dentistry. This study is justified by the importance of the association of postural evaluation methods mentioned above and the lack of studies with this approach.

Considering the need to establish healthy postural habits from the beginning of the professional training of the dentist and the possible positive effects of the implementation of magnification during preclinical training in these habits it is important to obtain scientific evidence to prove these effects. Therefore, the aim of this study was to assess the effects of using different magnification systems on the angular deviations from the neutral positions of the neck and trunk and the activities of the muscles of the upper back and neck during preclinical cavity preparation.

The null hypothesis is that the use of different magnification systems has no effect on the angular deviations from the neutral positions of the neck and trunk and the activities of the muscles of the upper back and neck during preclinical cavity preparation. The alternative hypothesis is that the use of different magnification systems has an effect on the angular deviations from the neutral positions of the neck and trunk and the activities of the muscles of the upper back and neck during preclinical cavity preparation.

## MATERIALS & METHODS

### Study design

This was an experimental laboratory study. The response variables were the angular deviations from the neutral positions of the neck and trunk, measured using the Software for Postural Assessment –SAPO software (version 0.69) (Laboratory for Biomechanics and Motor Control Federal University of ABC (UFABC), São Bernardo do Campo, São Paulo, Brazil. Available in: http://pesquisa.ufabc.edu.br/bmclab/sapo/), and the muscular activities of the bilateral descending trapezius, ascending trapezius, and sternocleidomastoid muscles, measured using surface EMG, while the subject performed a simulated clinical procedure, specifically, cavity preparation. The independent variables were the different magnification systems, at four levels (Simple, Galilean, and Keplerian loupes, with direct vision as the control) and the prepared teeth at four levels (teeth 16, 26, 36, and 46) (Fig. 1).

The sample unit was cavity preparation performed on each artificial tooth (maxillary right first molar, maxillary left first molar, mandibular left first molar and mandibular right first molar). Using data from a pilot study (mean and standard deviation), considering a power of 80% and a significance level of 5%, the minimum sample size was determined as
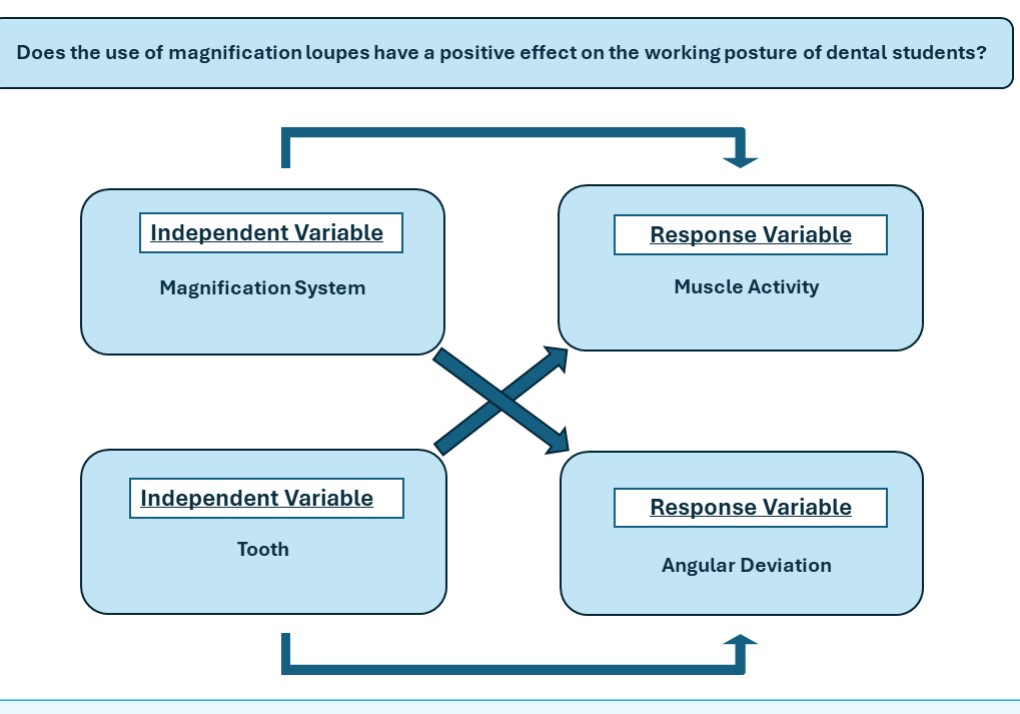

**Figure 1  Conceptual framework.**

10 procedures for each sample condition (Dimam software, Editora Guanabara, Koogan; Rio de Janeiro (RJ), Brazil).

This study was approved by the Research Ethics Committee of the School of Dentistry, São Paulo State University (UNESP), Araraquara, Brazil (CAAE Registry No. 50704921.1.0000.5416). Written consent was obtained from all participants.

## Magnification systems

The cavity preparations were performed either with direct vision, using a 3.5 × magnification Simple loupe (Bio-Art), a 3.5 × magnification Galilean system loupe (Ymarda Optical Instrument Factory, Nanjing, China), and a 4.0 × magnification Keplerian system loupe (Ymarda Optical Instrument Factory, Nanjing, China). Headband loupes were selected, because they can be worn over corrective or protective eyewear (Fig. 2).

## Cavity preparation

Class I cavity preparations for composite resin were performed on teeth 16 (maxillary right first molar), 26 (maxillary left first molar), 36 (mandibular left first molar) and 46 (mandibular right first molar) ($N = 160$).

The cavity preparations were performed following the quality criteria proposed by *Wajngarten et al. (2021)*. A #1012 round diamond bur was used at low speed and replaced after every 10 cavity preparations (Fig. 3). A dental mannequin with a phantom head (Manequins Odontológicos de Marília, MOM) with artificial resin teeth specifically designed for preclinical cavity preparation, were used in this study (Fig. 4). As the teeth were prepared, they were replaced with intact resin teeth for new preparations. The

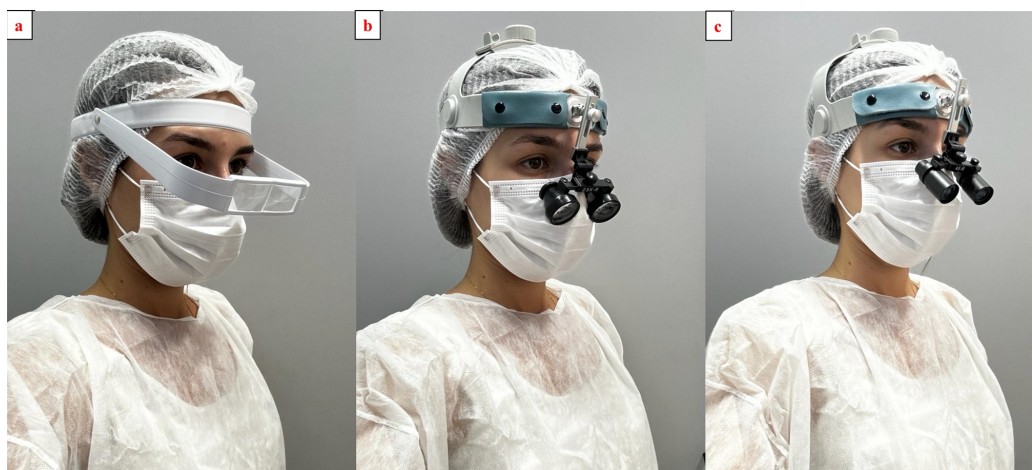

**Figure 2** Magnification systems: (A) simple loupe; (B) Galilean loupe; (C) Keplerian loupe.

dental phantom head was attached to the dental chair to simulate treatment in a clinical environment.

## Angular deviations

Participants were filmed during the entire procedure using a camera (GoPro Hero 4) positioned on a tripod placed 1m away and positioned to provide a lateral view of them. The angular deviations from the neutral positions of the neck and trunk were measured based on the RULA method (*McAtamney & Corlett, 1993*) by a trained and blinded researcher ($\rho_{neck} = 0,712$; $\rho_{trunk} = 0,935$) using the SAPO software (Fig. 5).

## Muscular activity

Surface EMG of the muscles of the upper back (the descending and ascending trapezius) and neck (sternocleidomastoid) muscles was performed to analyze muscle activity according to the protocol recommendations for non-invasive surface EMG assessment (the SENIAM guidelines) (*Hermens et al., 2000*). A portable electromyograph (MyoSystem-BrI; Datahominis Tecnologia, São Paulo, SP, Brazil) was used to record the EMG signals. The input impedance was 1010 $\Omega$/6 pf, polarization current input was $\pm 2$ nA, common mode rejection was 110 dB at 60 Hz, and gain was 20×. EMG signals were amplified 50× (total gain, 1000×), band-filtered (20 Hz–1 kHz), and sampled at a frequency of 2 kHz with 16-bit resolution.

The skin and the electrodes were cleaned using alcohol, in order to reduce skin impedance. The simple differential active bipolar electrodes were positioned over the belly of the muscle following the long axis of the muscle fibers on the right and left sternocleidomastoid, right and left descending trapezius, and right and left ascending trapezius muscles. The ground electrode was placed on the right wrist to ensure signal quality. After placing the electrodes and checking the electromyographic signal, the operator was instructed to initiate the cavity preparation (Fig. 6). EMG signals were recorded continuously for 120 s, during the complete preclinical cavity preparation

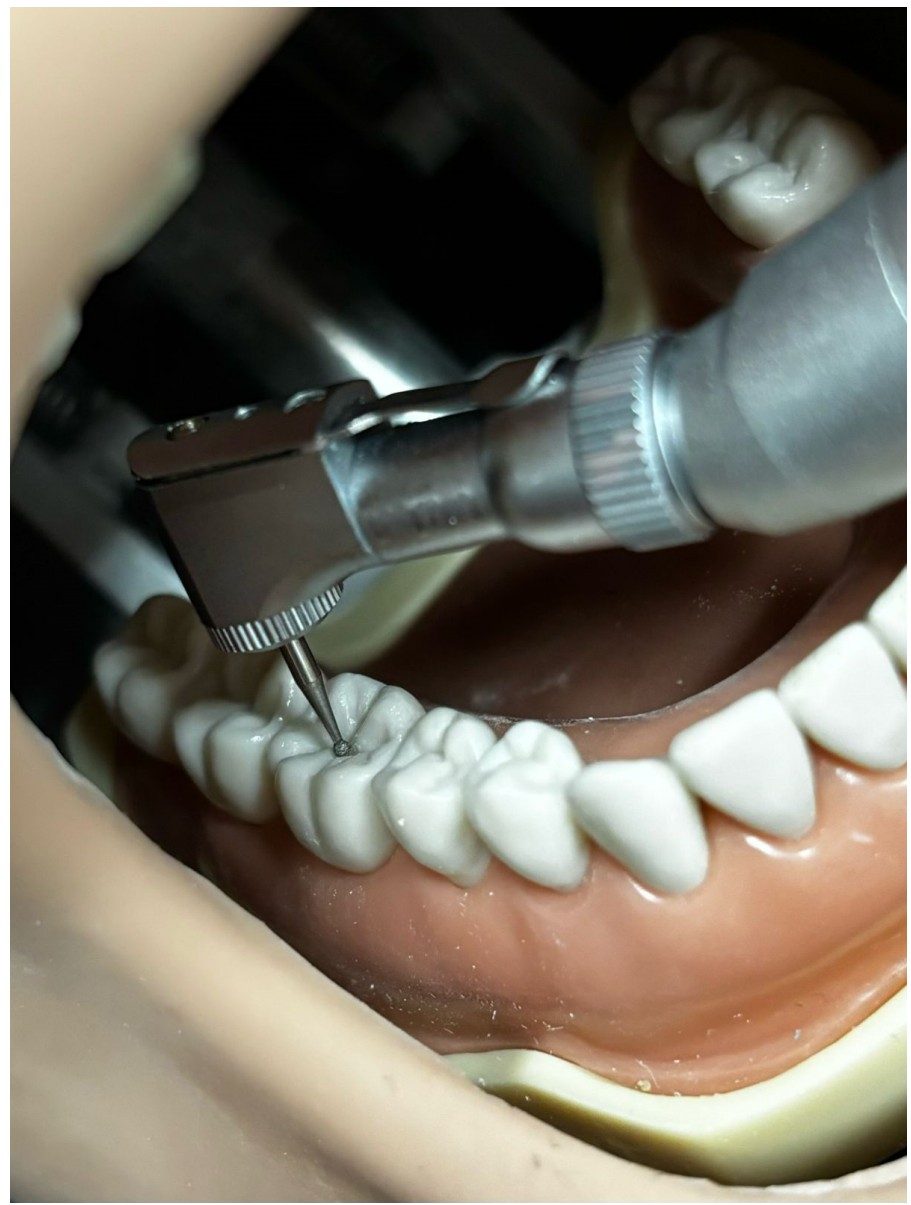

**Figure 3   Class I cavity preparation with a 1012 round diamond bur.**

procedure. This duration was determined based on the average time required for a Class I cavity preparation. The data were then stored, processed, and analyzed on a computer, using the MyoSystem software itself (*Milerad et al., 1991*).

To normalize the EMG signals, maximum voluntary contractions (MVC) in isometry against manual resistance were performed for 4 s for each muscle group investigated (*Haddad et al., 2012*). To do this, the operator was instructed to sit comfortably on a dental stool and perform the maximum possible contraction against manual resistance of the EMG laboratory technician. Subsequently, the operator took a 10 min break before the

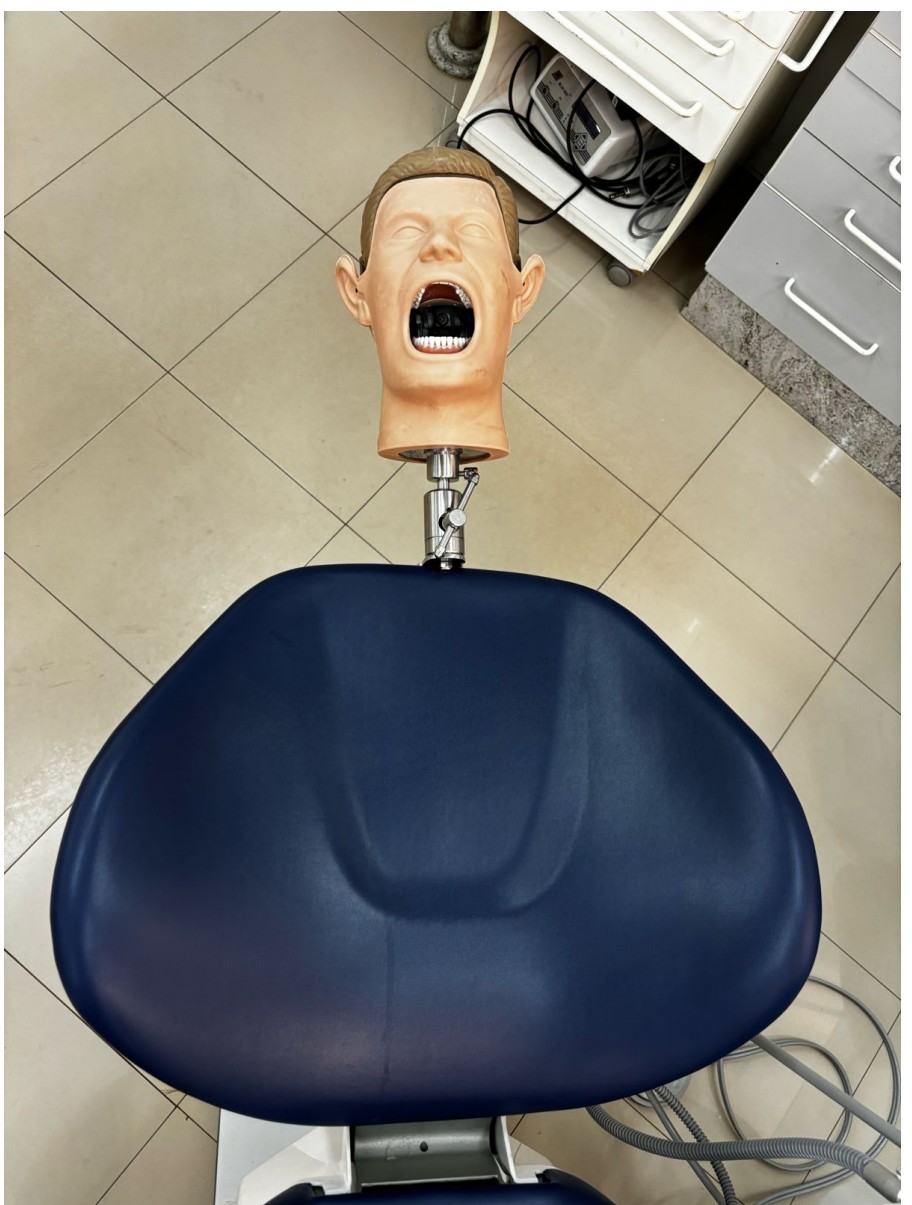

**Figure 4  Dental mannequin with the phantom head.**

next cavity preparation (*Pejcić et al., 2016*), in order to avoid muscle fatigue. The raw EMG data were filtered and rectified, and the root mean square (RMS) values were calculated.

## Statistical analysis

Descriptive statistical analysis was performed, and after verifying the assumptions of normality and homoscedasticity, two-way analysis of variance (ANOVA) and the Tukey and Games-Howell post-hoc tests were used to compare the data. The significance level adopted was 5%.

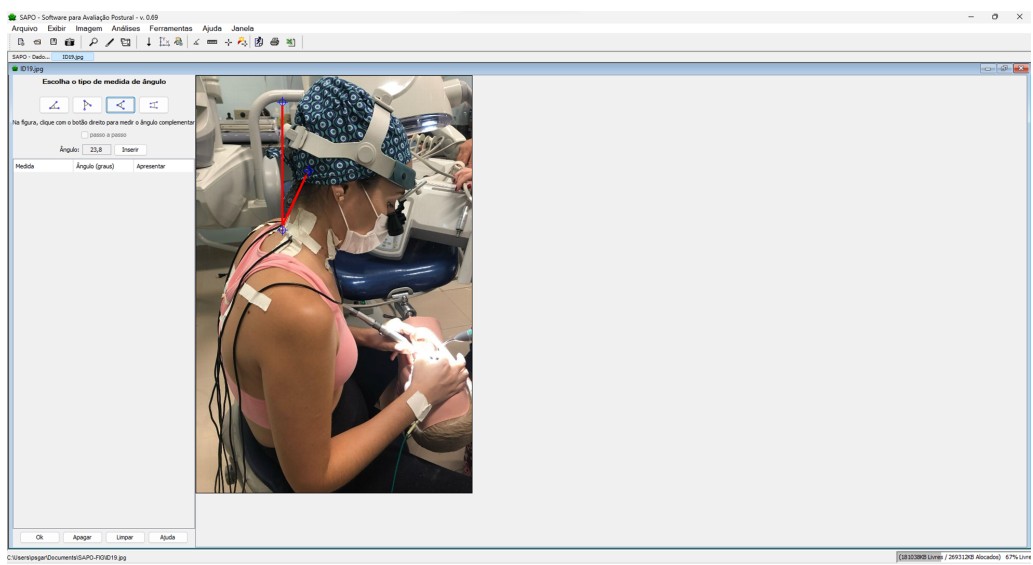

**Figure 5** Measuring angular deviation from the neutral position of the neck using the SAPO software.

**Table 1** Angular deviations from the neutral positions of the neck and trunk during cavity preparation of teeth 16, 26, 36, and 46, according to the magnification system used.

| Tooth | Magnification System | | | |
| --- | --- | --- | --- | --- |
| | Naked eye | Simple Loupe | Galilean Loupe | Keplerian Loupe |
| | Neck | | | |
| 16 | 35.93 ± 5.91 | 36.37 ± 8.16 | 31.66 ± 4.88 | 29.02 ± 3.90 |
| 26 | 35.90 ± 5.02 | 37.68 ± 5.99 | 30.97 ± 5.41 | 29.52 ± 3.47 |
| 36 | 32.44 ± 3.64 | 32.04 ± 5.09 | 26.49 ± 5.25 | 24.80 ± 3.99 |
| 46 | 33.71 ± 3.39 | 90.74 ± 3.15 | 29.79 ± 3.78 | 27.05 ± 3.69 |
| | Trunk | | | |
| 16 | 2.18 ± 0.48 | 2.05 ± 0.34 | 1.96 ± 0.39 | 2.06 ± 0.46 |
| 26 | 2.42 ± 0.63 | 2.19 ± 0.72 | 2.18 ± 0.27 | 2.10 ± 0.43 |
| 36 | 1.85 ± 0.37 | 1.97 ± 0.30 | 1.82 ± 0.25 | 1.75 ± 0.26 |
| 46 | 2.02 ± 0.57 | 2.09 ± 0.48 | 1.67 ± 0.28 | 1.93 ± 0.32 |

**Notes.**

Data are presented as the mean ± standard deviation values.

Two-way ANOVA results. Angular deviation of the neck: magnification system ($F = 19.733$, $p < 0.010$, $\pi = 1.000$), tooth ($F = 8.561$, $p < 0.01$, $\pi = 0.993$), and magnification system × tooth ($F = 0.558$, $p = 0.829$, $\pi = 0.267$). Angular deviation of the trunk: magnification system ($F = 2.050$, $p = 0.109$, $\pi = 0.517$), tooth ($F = 5.791$, $p = 0.010$, $\pi = 0.947$), and magnification system × tooth ($F = 0.555$, $p = 0.832$, $\pi = 0.265$).

## RESULTS

Table 1 presents the mean and standard deviation values of the angular deviations from the neutral positions of the neck and trunk during cavity preparation for teeth 16, 26, 36, and 46, according to the magnification system used. The two-way ANOVA results are also summarized.

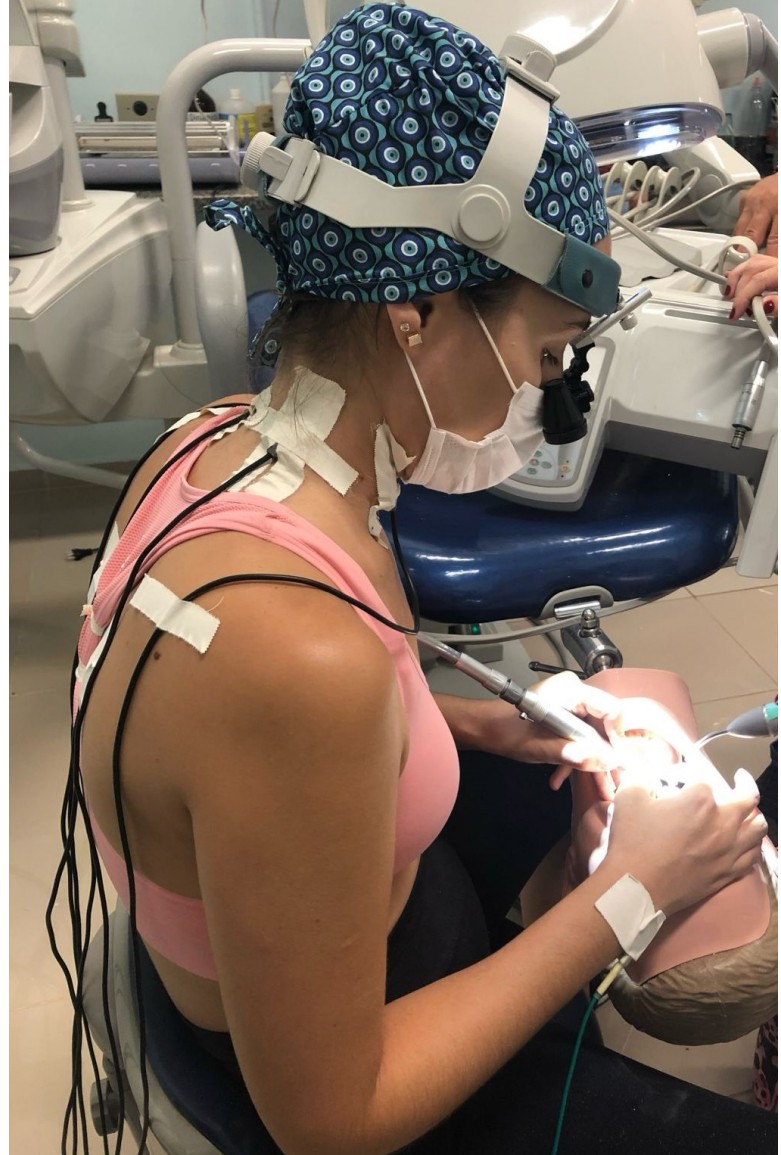

**Figure 6** **Electromyographic data collection during cavity preparations.**

With regard to the angular deviation of the neck, there was no significant interaction between the factors "magnification system" and "tooth" ($p = 0.829$). Statistical significance was verified for the factors considered separately ($p_{\text{magnification system}} < 0.010$; $p_{\text{tooth}} < 0.010$), and corresponding 95% confidence intervals (CIs) for angular deviations from the neutral neck position were derived (Fig. 7).

Angular deviations from the neutral position of the neck were significantly higher during cavity preparation for teeth 16 ($CI_{95\%} = 31.211$–$35.278$) and 26 ($CI_{95\%} = 31.653$–$35.382$), regardless of the magnification system used. Angular deviation from the neutral position of the neck was significantly higher during cavity preparations performed with the naked eye

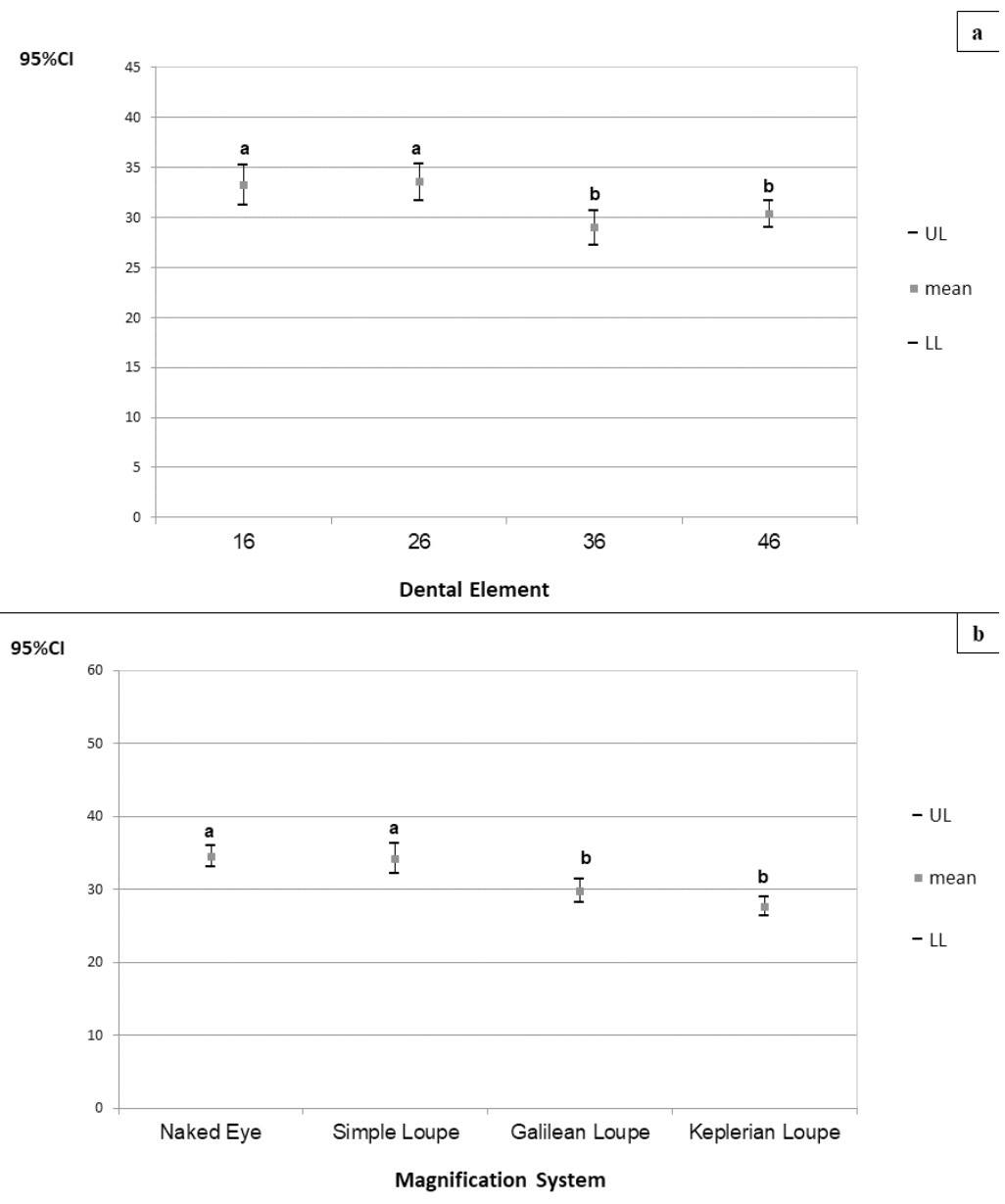

**Figure 7** (A–B) 95% confidence intervals (CIs) of the angular deviations from the neutral position of the neck according to the magnification systems (A) and prepared teeth (B), respectively. Tukey's *post-hoc* test; (A–B) equal letters represent statistical similarity.

(CI$_{95\%}$ =33.027–35.963) and using the simple loupe (CI$_{95\%}$ =32.211–36.203), regardless of the prepared tooth. The angular deviations of the neck for preparations performed using the Galilean (CI$_{95\%}$ =28.128–31.327) and Keplerian loupes (CI$_{95\%}$ =26.317–28.878) did not differ significantly.

With regard to the angular deviation from the neutral position of the trunk, there was no significant interaction between the factors "magnification system" and "tooth" ($p = 0.832$).

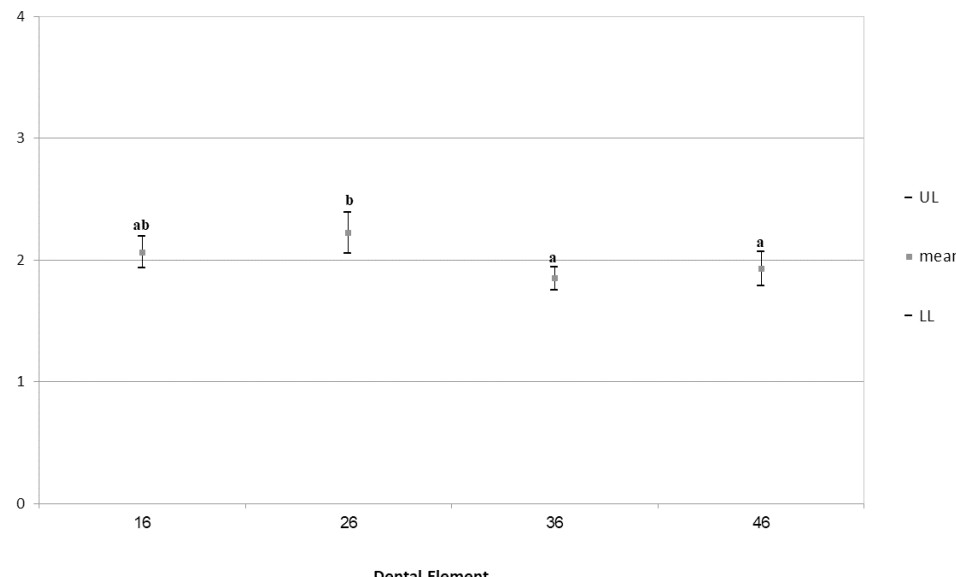

**Figure 8** **95% confidence intervals (CIs) of the angular deviations from the neutral position of the trunk according to prepared tooth.** Tukey's *post-hoc* test; (A–B) equal letters represent statistical similarity.

Only the factor "tooth" was statistically significant ($p = 0.010$), and corresponding 95% CIs were derived for the angular deviations from the neutral position of the trunk (Fig. 8).

The angular deviation of the trunk was significantly higher during cavity preparations on tooth 26 ($CI_{95\%} = 2.055-2.390$), regardless of the magnification system used.

Table 2 presents the mean and standard deviation values of the normalized EMG data for the right and left sternocleidomastoid, right and left descending trapezius, and right and left ascending trapezius muscles during cavity preparations of teeth 16, 26, 36, and 46, according to the magnification system used. The two-way ANOVA results are also summarized.

Table 2: Normalized EMG values of the right and left sternocleidomastoid, right and left descending trapezius, and right and left ascending trapezius muscles during cavity preparation of teeth 16, 26, 36, and 46, according to the magnification system used.

For the left sternocleidomastoid, right and left descending trapezius, and left ascending trapezius muscles, there was a significant interaction between the factors "magnification system" and "tooth" (p<0.010−0.013), For the left sternocleidomastoid, higher muscle activity was observed while working on tooth 46 with the Galilean loupe, which was significantly different from observed for tooth 36. For the Keplerian loupe, higher muscle activity was also observed during work on tooth 46, which was significantly different from that of the other prepared teeth.

For the right descending trapezius, while working on teeth 16 and 26, higher muscle activity was observed when using the Simple and Keplerian loupes. For teeth 36 and 46,

**Table 2  Normalized EMG values.**

| Tooth | Magnification System | | | |
|---|---|---|---|---|
| | **Naked eye** | **Simple Loupe** | **Galilean Loupe** | **Keplerian Loupe** |
| | **Right Sternocleidomastoid** | | | |
| 16 | 7.33 ± 1.47 | 6.95 ± 0.83 | 5.72 ± 1.37 | 6.91 ± 0.86 |
| 26 | 6.00 ± 1.38 | 6.89 ± 1.14 | 5.97 ± 1.22 | 6.67 ± 1.17 |
| 36 | 6.30 ± 2.35 | 6.18 ± 0.69 | 5.79 ± 1.21 | 6.75 ± 0.81 |
| 46 | 6.14 ± 1.63 | 5.93 ± 0.62 | 5.43 ± 1.46 | 5.98 ± 0.57 |
| | **Left Sternocleidomastoid** | | | |
| 16 | 2.80 ± 0.63**Aa** | 3.34 ± 0.75**Aa** | 3.27 ± 0.45**Aa** | 3.41 ± 0.61**Aa** |
| 26 | 3.25 ± 0.52**Aa** | 3.07 ± 1.13**Aa** | 3.37 ± 0.57**Aa** | 3.47 ± 0.62**Aa** |
| 36 | 2.68 ± 0.52**Aa** | 3.12 ± 0.57**Aa** | 2.91 ± 0.37**Aab** | 3.10 ± 0.77**Aa** |
| 46 | 4.43 ± 1.67**Aa** | 4.07 ± 1.46**Aa** | 4.57 ± 1.06**Aac** | 6.11 ± 0.81**Ab** |
| | **Right descending trapezius** | | | |
| 16 | 30.84 ± 2.50**Aa** | 66.13 ± 13.81**Ba** | 34.56 ± 9.06**Aa** | 61.57 ± 8.35**Ba** |
| 26 | 26.69 ± 4.71**Aab** | 42.60 ± 7.29**Bb** | 23.02 ± 5.46**Aa** | 36.68 ± 3.64**Bb** |
| 36 | 25.72 ± 3.13**Ab** | 44.29 ± 4.19**Bb** | 23.95 ± 3.69**Aa** | 27.79 ± 4.44**Ac** |
| 46 | 25.73 ± 5.53**Aab** | 39.95 ± 6.42**Bb** | 25.65 ± 6.78**Aa** | 34.38 ± 4.69**ABbc** |
| | **Left descending trapezius** | | | |
| 16 | 36.49 ± 8.85**Aa** | 48.22 ± 5.14**ABa** | 41.12 ± 12.72**Aa** | 36.52 ± 4.50**ACa** |
| 26 | 19.93 ± 1.50**Ab** | 22.78 ± 1.97**ABb** | 20.96 ± 2.45**Ab** | 18.32 ± 2.57**ACb** |
| 36 | 11.46 ± 2.01**Ac** | 23.96 ± 2.93**Bb** | 12.52 ± 3.70**Ac** | 15.46 ± 2.49**Ab** |
| 46 | 15.96 ± 6.21**Abc** | 25.32 ± 3.51**Bb** | 13.65 ± 5.81**Abc** | 16.84 ± 3.61**Ab** |
| | **Right ascending trapezius** | | | |
| 16 | 10.26 ± 2.53 | 12.39 ± 1.44 | 11.41 ± 3.72 | 13.76 ± 3.22 |
| 26 | 8.39 ± 2.16 | 13.74 ± 3.90 | 9.61 ± 3.81 | 11.63 ± 3.24 |
| 36 | 12.77 ± 2.91 | 17.02 ± 3.83 | 14.38 ± 3.75 | 14.89 ± 1.90 |
| 46 | 14.11 ± 5.30 | 16.43 ± 3.11 | 13.22 ± 2.75 | 20.38 ± 3.62 |
| | **Left ascending trapezius** | | | |
| 16 | 13.93 ± 3.65**Aa** | 16.87 ± 3.84**Aa** | 11.85 ± 4.41**Aa** | 15.80 ± 4.68**Aab** |
| 26 | 16.90 ± 4.91**Aa** | 26.31 ± 4.05**Bb** | 14.84 ± 6.08**Aa** | 26.12 ± 8.66**ABa** |
| 36 | 14.97 ± 2.95**Aa** | 18.11 ± 3.21**Aa** | 12.69 ± 3.25**Aa** | 13.52 ± 1.75**Ab** |
| 46 | 17.05 ± 4.57**Aa** | 20.66 ± 5.48**Aab** | 16.20 ± 3.84**Aa** | 16.53 ± 1.53**Aa** |

**Notes.**

Data are presented as the mean ± standard deviation values.

Two-way ANOVA results. Right sternocleidomastoid: magnification system ($F = 3.895$, $p = 0.008$, $\pi = 0.818$), tooth ($F = 3.195$, $p = 0.023$, $\pi = 0.728$), and magnification system × tooth ($F = 0.727$, $p = 0.684$, $\pi = 0.349$). Left Sternocleidomastoid: magnification system ($F = 5.664$, $p = 0.077$, $\pi = 0.942$), tooth ($F = 37.805$, $p < 0.010$, $\pi = 1.000$), and magnification system × tooth ($F = 2.443$, $p = 0.013$, $\pi = 0.915$). Right descending trapezius: magnification system ($F = 104.624$, $p < 0.010$, $\pi = 1.000$), tooth ($F = 68.956$, $p < 0.010$, $\pi = 1.000$), and magnification system × tooth ($F = 8.813$, $p < 0.010$, $\pi = 1.000$). Left descending trapezius: magnification system ($F = 26.653$, $p < 0.010$, $\pi = 1.000$), tooth ($F = 191.053$, $p < 0.010$, $\pi = 1.000$), and magnification system × tooth ($F = 2.405$, $p < 0.010$, $\pi = 0.910$). Right ascending trapezius: magnification system ($F = 13.242$, $p < 0.010$, $\pi = 1.000$), tooth ($F = 21.066$, $p < 0.010$, $\pi = 1.000$), and magnification system × tooth ($F = 1.925$, $p = 0.053$, $\pi = 0.821$). Left ascending trapezius: magnification system ($F = 16.158$, $p < 0.010$, $\pi = 1.000$), tooth ($F = 17.956$, $p < 0.010$, $\pi = 1.000$), and magnification system × tooth ($F = 3.073$, $p < 0.010$, $\pi = 0.969$). Games-Howell post-hoc test. * Lowercase letters represent rows, and uppercase letters represent columns. The same letters indicate statistical similarities.

Bold text indicates statistical similarity with the same letters indicating statistical similarity and different letters indicating statistical difference.

the highest activity was observed with the use of the Simple loupe, which did not differ significantly from that of the Keplerian loupe for tooth 46.

For the left descending trapezius muscle, higher muscle activity was observed for all teeth when using the Simple loupe, which differed significantly from those observed with the use of the other magnification systems when preparing teeth 36 and 46.

For the left ascending trapezius, there were no significant differences in muscle activity when using the different magnification systems for preparing teeth 16, 36, and 46. For tooth 26, the highest muscle activity occurred during the use of the Simple loupe, and it was similar to that observed using the Keplerian loupe. For the naked eye and the Galilean loupe, there was no significant difference in muscle activity during the preparation of the different teeth.

For the right sternocleidomastoid and right ascending trapezius muscles, there was a significant difference only for the factors when considered separately ($p_{\text{magnification system}}$ = 0.008 and $p_{\text{magnification system}}$<0.010, respectively; $p_{\text{tooth}}$ = 0.023 and $p_{\text{tooth}}$<0.010, respectively), but no interaction of the factors "magnification system" and "tooth" ($p = 0.684$ and $p = 0.053$, respectively). The corresponding 95% CIs are presented in Figs. 9 and 10.

It can be seen that the activity of the right sternocleidomastoid muscle during cavity preparation on tooth 16 ($\text{CI}_{95\%}$ =6.325−7.127) was higher and significantly different from that for tooth 46 ($\text{CI}_{95\%}$ =5.505−6.232). Moreover, the activities of this muscle during cavity preparation using the Galilean loupe were significantly lower than those when using the Simple and Keplerian loupes.

The right ascending trapezius muscle showed higher muscle activity during cavity preparations on teeth 36 ($\text{CI}_{95\%}$ =13.688–15.844) and 46 ($\text{CI}_{95\%}$ =14.587–17.487), and these were significantly differing from those for teeth 16 ($\text{CI}_{95\%}$ =10.999–12.908) and 26 ($\text{CI}_{95\%}$ =9.641–12.040). Moreover, its activities were higher and significantly different when using the Simple ($\text{CI}_{95\%}$ =13.752–16.043) and Keplerian loupes ($\text{CI}_{95\%}$ =13.782–16.545).

## DISCUSSION

In this study it was observed that using the Galilean and Keplerian loupes resulted in less angular deviation of the neck and, in general, the Galilean loupe system resulted in lower muscle activity in the evaluated regions. Similar results have been reported previously. *Wajngarten & Garcia (2019)* and *Wajngarten et al. (2021)* observed lower neck angulation in dental students while using the Galilean and Keplerian loupes compared to using the naked eye and the Simple loupe. *Pazos et al. (2022)* observed less angular deviation from the neutral neck position while working with the Galilean loupe, and *Kamal et al. (2020)* reported positive effects of using magnification loupes on the working posture of the neck and trunk of dental students at the preclinical level. Regarding muscle activity, *Bud et al. (2021)* have reported that the use of the Galilean loupe and the operative microscope improved the working posture of dental students, helping to maintain the correct positions of the head, neck, and shoulders. *García-Vidal et al. (2019)* evaluated the effect of using the Galilean loupe and/or an ergonomic stool on muscle activity in the upper back region and reported that both separate and combined use of the magnification loupe and dental stool resulted in significant reduction in muscle activity in the evaluated region.

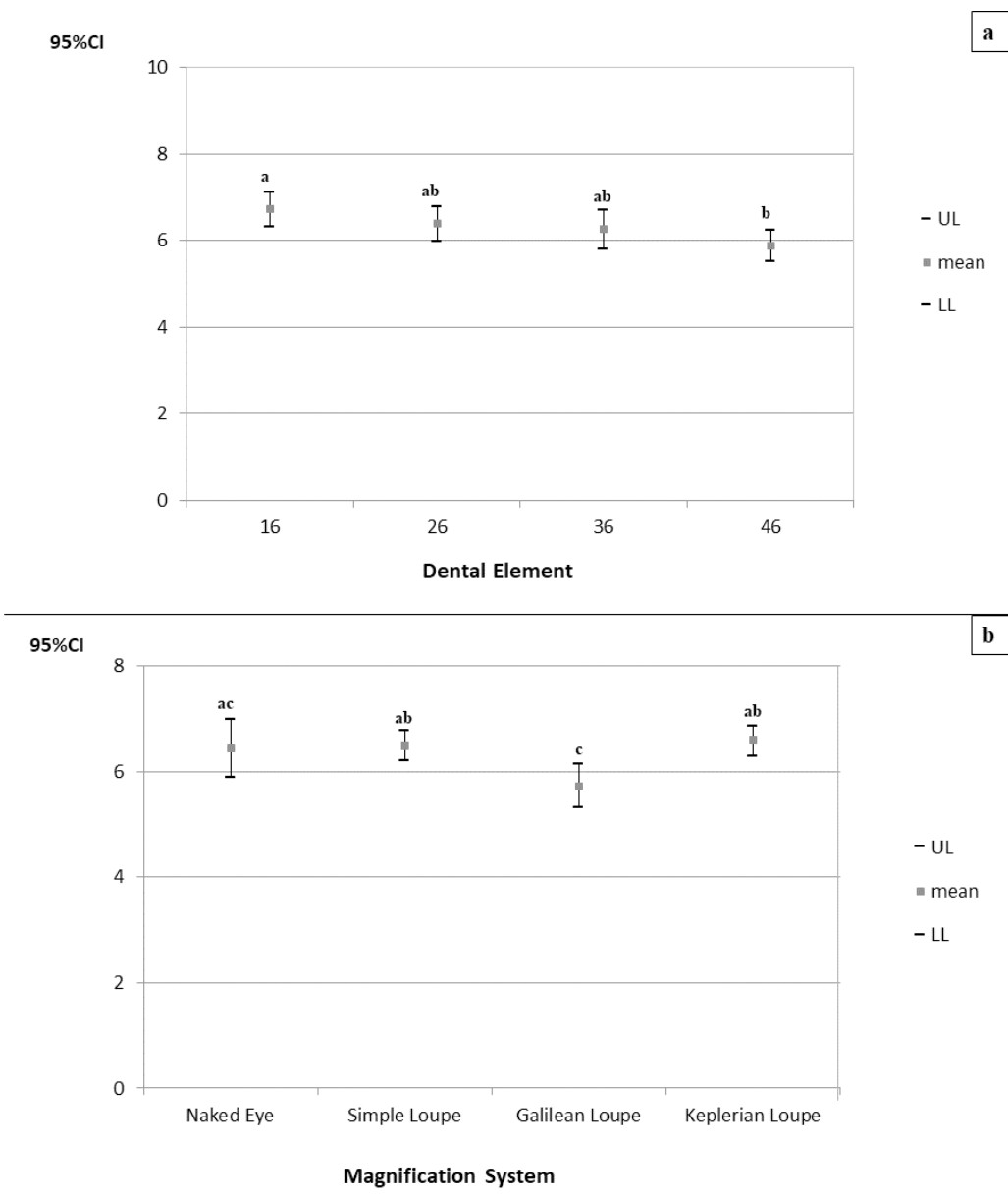

**Figure 9** (A–B) 95% confidence intervals (CIs) of muscle activities of the right sternocleidomastoid muscle according to the prepared tooth (a) and the magnification system (b), respectively. Games-Howell's *post-hoc* test; (A–B) equal letters represent statistical similarity.

These observations may be explained by the improvement in the visualization of the working field and the adequate focal length provided by the optical complexity of these systems, which are expected to allow for a better visualization angle (*Kamal et al., 2020*; *Wajngarten et al., 2021*; *Pazos et al., 2022*). Additionally, the use of these loupes allows greater visual acuity at an adequate distance from the operative field (*Eggmann et al., 2022*), enabling a more neutral working posture (*Eichenberger et al., 2011*; *Kamal et al., 2020*) and

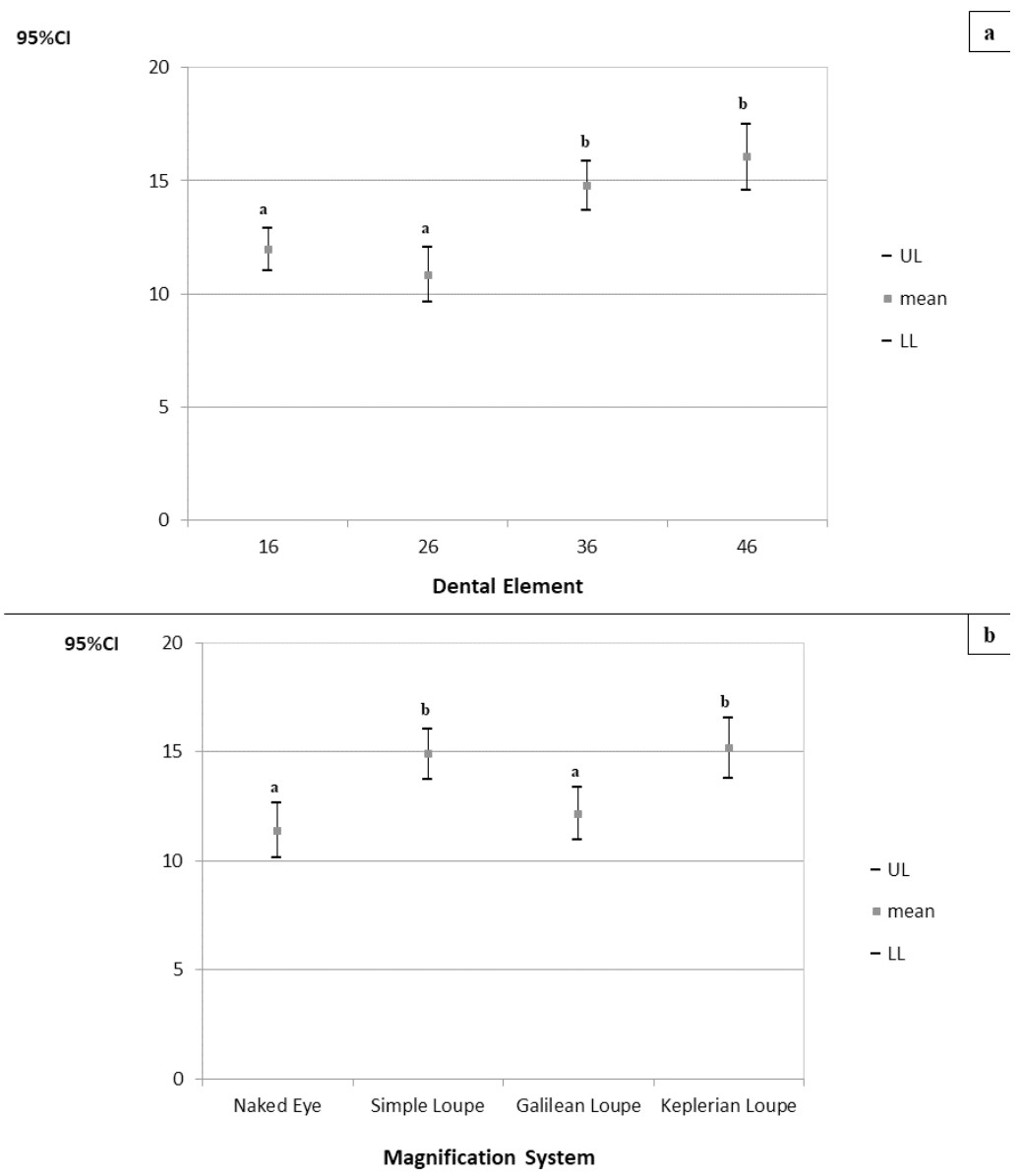

**Figure 10** **(A–B) 95% confidence intervals (CIs) of the muscle activities of the right ascending trapezius muscle according to the prepared tooth (a), and the magnification system (b), respectively.** Games-Howell's *post-hoc* test; (A–B) equal letters represent statistical similarity.

consequently, less muscle activity. Conversely, although the Simple loupe provides good magnification in the operative field, it is associated with limited working distance and depth of field (*Shanelec, 1992*; *Eichenberger et al., 2011*). This requires the operator to be closer to the working field for improved focus, which consequently compromises the working posture and results in increased angular deviation from the neutral position of the neck (*Eichenberger et al., 2011*; *Wajngarten & Garcia, 2019*).

According to *McAtamney & Corlett (1993)*, a lower neck angle is associated with a lower risk of developing musculoskeletal disorders, although the lowest risk scores were related to angular deviations of up to 10° (*McAtamney & Corlett, 1993*). Therefore, even though the Galilean and Keplerian loupes resulted in reduced angular deviation of the neck, they failed to result in acceptable angular deviation. This may have been because the declination angle of the magnification loupes must be optimized so that the operator can find an ideal balance between eye strain and neck angulation (*Rucker et al., 1999*; *Pazos et al., 2022*). In this study, semi-adjustable loupes were used, which allow manual adjustments of the declination angle and interpupillary distance in each use. It is possible that this lack of personalization of the loupes influenced neck inclination.

Regarding tooth location, angular deviation from the neutral position of the neck was higher during cavity preparation of teeth 16 and 26, but from the neutral position of the trunk only for tooth 26. It is possible that these results are also related to the declination angle of the loupes. As the adjustment of this angulation can differ for both arches, it is possible that this angle was not properly adjusted for the upper arch, making a larger inclination of the neck necessary for adequate visualization of the teeth in this arch (*Rucker et al., 1999*). Furthermore, the fact that tooth 26 is on the opposite side of the right-handed operator may have influenced the greater inclination of both the trunk and neck during work in this region. Another possible reason is the difficulty faced by dental students in properly positioning their patients in the dental chair. During the training phase, it is common for students to feel insecure about patient care and about positioning the dental chair in a way that makes the patient uncomfortable (*Presoto, Wajngarten & Garcia, 2016*; *Wajngarten et al, 2021*). Inadequate positioning of the dental chair may require a greater inclination of the neck for adequate visualization of the operative field.

*Pazos et al. (2022)* also observed a higher angular deviation from the neutral position of the neck during restorative procedures on the teeth of the upper arch. However, *Kamal et al. (2020)* and *Wajngarten et al. (2021)* found that the working posture of dental students was negatively affected while working on lower arch teeth.

Taken together, the findings of previous studies and the results obtained in this study indicate that the Galilean loupe is a good option for the dental teaching environment, mainly in the pre-clinical training phase. This could be explained based on the characteristics of this system, such as adequate working distance, intermediate increase in the working field, small size, and low weight (*Shanelec, 1992*; *James & Gilmour, 2010*; *Eichenberger et al., 2013*; *Wajngarten & Garcia, 2018*; *Pazos et al., 2020*; *Pazos et al., 2022*). Furthermore, the process of adapting its use is simpler. The Keplerian loupe system is characterized by greater magnification power, a smaller field of view, greater working distance, and better visual acuity and depth of field (*Shanelec, 1992*; *Eichenberger et al., 2011*). The last feature causes visual focus to be easily lost even with small operator movements (*Wajngarten & Garcia, 2019*), which can hinder the operator's work and the generation of more muscle tension in order to prevent small head movements. Additionally, the Keplerian loupe is larger and heavier, which can lead to higher muscle activity to compensate for this weight.

A limitation of this study is that postural analysis was performed only during cavity preparation. This procedure was chosen because the literature is more focused in the

relationship between magnification and endodontics, and there are few studies related to restorative procedures (*Eggmann et al., 2022*). Despite that, many other procedures are also common in clinical practice, which the effect of the use of magnification could be more substantial. Therefore, future studies may consider other clinical procedures. Furthermore, no training was given prior to using the magnification loupes, which may have interfered with the results, as the learning curve for this device can be long and challenging (*Eggmann et al., 2022*). Nevertheless, this study also has several strengths. First, continuous electromyographic measurements were obtained during the entire procedure (duration: approximately 120 s), which allowed us to obtain more reliable data in the context of the actual dental clinical setting. Additionally, bilateral EMG recordings were obtained, which allowed a comprehensive assessment of the targeted muscles. Finally, the combination of surface EMG and angular deviation measurements provided additional information that led to a more detailed analysis of the effects of magnification on working posture.

In general, the results of this study corroborate those of previous studies (*Eichenberger et al., 2011*; *Carpentier et al., 2019*; *Kamal et al., 2020*; *Wajngarten & Garcia, 2019*; *Pazos et al., 2020*; *Pazos et al., 2022*) that support the recommendations for the use of magnification loupes to prevent the development of musculoskeletal disorders as well as the implementation and training still in the training phase (*Carpentier et al., 2019*; *Kamal et al., 2020*; *Braga et al., 2021*; *Pazos et al., 2022*).

The implementation of a training program for the use of magnification loupes at this stage is justified by the fact that students are still inexperienced in dental care, which makes postural changes easier because they do not have established deleterious habits (*Carpentier et al., 2019*). Furthermore, taking into account the fact that no training for the use of loupes had been applied prior to this study, it is possible to assume that the improvement in working posture attested by less muscle activity and angular deviation from the neutral neck position occurred from spontaneous way. Thus, training using magnification loupes during the pre-clinical training phase could bring even more postural benefits to this population.

Furthermore, training and use of magnification since the professional training phase can be a strategy to facilitate the clinical implementation of magnification in dentistry, since if students acquire magnification loupes and create the habit of working with them during dental school, they will most likely continue to do so during the exercise of their profession. In the long term, this habit could directly impact the longevity of dentists in their careers, due to the reduction in the development of musculoskeletal disorders (*Carpentier et al., 2019*; *Kamal et al., 2020*; *Braga et al., 2021*; *Pazos et al., 2022*), in addition to bringing clinical benefits to patients through improvements in diagnostic capacity and the quality of treatment provided (*Braga et al., 2021*; *Eggmann et al., 2022*).

## CONCLUSIONS

It can be concluded that Galilean loupe resulted in lower muscle activity in the neck and back regions and that the Galilean and Keplerian loupes resulted in less angular deviations

of the neck and trunk during cavity preparation. It may suggest that Galilean loupe can be an adequate system for implementation during the pre-clinical training phase.

### Funding
This work was supported by São Paulo Research Foundation (FAPESP) Grant #2019/25528-4. This study was financed by the Coordenação de Aperfeiçoamento de Pessoal de Nível Superior - Brasil (CAPES) - Finance Code 001. The funders had no role in study design, data collection and analysis, decision to publish, or preparation of the manuscript.

### Grant Disclosures
The following grant information was disclosed by the authors:
São Paulo Research Foundation (FAPESP): #2019/25528-4.
Coordenação de Aperfeiçoamento de Pessoal de Nível Superior - Brasil (CAPES) - Finance Code 001.

### Competing Interests
The authors declare there are no competing interests.

### Author Contributions
- Júlia Margato Pazos performed the experiments, analyzed the data, prepared figures and/or tables, authored or reviewed drafts of the article, and approved the final draft.
- Ana Flávia Ribeiro Monteiro Fernandes performed the experiments, prepared figures and/or tables, and approved the final draft.
- Edson Donizetti Verri performed the experiments, prepared figures and/or tables, and approved the final draft.
- Guilherme Gallo Costa Gomes performed the experiments, prepared figures and/or tables, and approved the final draft.
- Simone Cecílio Hallak Regalo conceived and designed the experiments, authored or reviewed drafts of the article, and approved the final draft.
- Patricia Petromilli Nordi Sasso Garcia conceived and designed the experiments, analyzed the data, authored or reviewed drafts of the article, and approved the final draft.

### Human Ethics
The following information was supplied relating to ethical approvals (*i.e.*, approving body and any reference numbers):
   This study was approved by the Research Ethics Committee of the School of Dentistry, São Paulo State University (UNESP), Araraquara, Brazil (CAAE Registry No. 50704921.1.0000.5416). Written consent was obtained from all participants.

### Data Availability
   The raw data is available in the Supplemental File.

## Supplemental Information

Supplemental information for this article can be found online at http://dx.doi.org/10.7717/peerj.17188#supplemental-information.

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
