# Peer review of "Magnification in preclinical procedures: effect on muscle activity and angular deviations of the neck and trunk"

_PeerJ, doi:10.7717/peerj.17188_

## Round 0.1 · original submission · Major Revisions

Please find the comments of the esteemed reviewers.

·

Basic reporting

In Abstract the authors wrote:
Galilean loupe resulted in lower muscle activity in the neck and back regions and that the Galilean and Keplerian loupes resulted in less angular deviations of the neck and trunk during cavity preparation.

In full manuscript the authors wrote:
Galilean system generally promotes less angular
deviation from the neutral neck position and less muscle activity in the evaluated regions

The conclusion in Abstract is quite different with Conclusion in full paper. Which one is correct???

Experimental design

In this study, there is no data on the age and gender of participants, which may affect the results of this study.

Validity of the findings

There was no conclusion about effect of different magnification system on angular deviation of the trunk since the purpose of this study was to assess the effects of using different magnification systems on the angular deviations from the neutral positions of the neck and trunk

Additional comments

-

Reviewer 2 ·

Basic reporting

o Title is clear and concise, accurately reflecting the research question.
o Abstract provides a good overview of the study aims, methods, results, and conclusions.
o Keywords are appropriate and facilitate literature search

Experimental design

2. EXPERIMENTAL DESIGN:
• Strengths:
o Clear study design: Randomized controlled trial with appropriate control group (direct vision).
o Adequate sample size based on pilot study.
o Blinding: Not mentioned, but blinding the researcher analyzing EMG and angular deviations would strengthen the design.
o Predefined outcome measures: Angular deviations and muscle activity.
o Standardization of procedures: Identical cavity preparation procedure, controlled environment.
• Weaknesses:
o Limited scope: Only assessed cavity preparation, not other dental procedures.
o Potential bias: Participants were not trained on loupe use beforehand, introducing a learning curve effect.
o Semi-adjustable loupes: Individual adjustments might not have been optimal for all participants, affecting posture data.

Validity of the findings

• Strengths:
o Objective outcome measures: EMG and angular deviation measurements provide quantitative data.
o Statistical analysis: Appropriate two-way ANOVA and post-hoc tests used to analyze data.
o Replicability: Detailed methods allow for replication of the study.
• Weaknesses:
o Generalizability: Results apply to preclinical students performing cavity preparation, may not generalize to experienced dentists or other procedures.
o Interpretation: Muscle activity data requires careful interpretation due to potential confounding factors like learning curve or fatigue.
o Lack of training: No data on how training in loupe use could influence results.

Additional comments

DETAILED REVIEW

Strengths:
• Clear writing and well-structured: The introduction provides a good background and context for the research question. The methods section is detailed, and the results are clearly presented.
• Original research: This study addresses a relevant topic in dentistry and fills a gap in the existing literature by using both EMG and angular deviation measurements to assess the impact of magnification on posture and muscle activity.
• Rigorous methods: The study design is appropriate, and the sample size is adequate. The statistical analysis is sound and appropriate for the data.
• Valuable findings: The results provide important insights into the effects of different magnification systems on posture and muscle activity. The findings are generally supportive of the use of magnification loupes, particularly the Galilean system, to improve posture and reduce the risk of musculoskeletal disorders.
Weaknesses:
• Limited scope: The study only focused on cavity preparation, which is a common procedure but does not represent the full range of tasks performed by dentists. Future studies could investigate the effects of magnification on other procedures.
• Potential limitations of loupe adjustment: The study used semi-adjustable loupes, which may not have been optimally adjusted for all participants. This could explain why the participants still had some neck angulation even with the Galilean and Keplerian loupes.
• Lack of training for loupe use: The participants did not receive prior training on how to use loupes. It is possible that training could further improve their posture and reduce muscle activity.

Additional comments for each section:
Introduction:
• The authors could briefly mention other interventions that have been shown to improve posture in dentistry, such as ergonomic chairs and work stools.
• The introduction could be strengthened by stating the specific hypotheses of the study.
Methods:
• The authors should provide more information about the dental phantom head and mannequin used in the study.
• The description of the EMG recording and analysis procedures could be more detailed.
Results:
• The results could be presented more visually with the help of sketch diagrams
• The authors could discuss the potential reasons for the higher angular deviation from the neutral position of the trunk during cavity preparation on tooth 26.
Discussion:
• The discussion could be expanded to include a more detailed comparison of the findings of this study with previous research.
• The authors could discuss the limitations of the study in more detail, such as the lack of training for loupe use and the use of semi-adjustable loupes.
• The authors could provide specific recommendations for the use of magnification loupes in the pre-clinical training phase.
Conclusion:
• The conclusion should summarize the main findings of the study.

·

Basic reporting

The manuscript under review attempts to evaluate the effect of muscle activity and angular deviations of the neck and trunk on magnification in preclinical procedures. In general, the manuscript captures details of the study design and implementation of the project. The manuscript's sections are well written and concluded, but the methodology needs corrections and revision. Although the limitations are mentioned, they must be presented in the manuscript. The study is of sound design and clear practical interest.

The abstract is structured and well-written
Introduction:
1. Kindly write briefly on the problem statement with recent references
2. Kindly provide the justification for the current study
3. Mention the recent literature and methods concerning the current study

Experimental design

Materials & Methods:
1. SAPO software: kindly provide the complete form
2. (teeth 16, 26, 36, and 46): kindly specify as maxillary and mandibular first molars
3. artificial tooth: kindly specify
4. The sample size was determined as 10 procedures for each sample condition: write briefly how the calculation was carried out. Using?
5. Kindly provide an illustration for the conceptual framework.
6. Figures or images of each step Magnification systems, Cavity preparation, Angular deviations, Muscular activity

Validity of the findings

Results:
1. Figures need to be explained clearly
2. The study's outcome has no significant difference and is almost objective; kindly present it well.
3. Kindly provide figures showing the outcome of the current study.
4. The figures provided show the statistical analysis, and it can be merged.

Additional comments

Discussion:
1. The first paragraph is the objective and outcome of the study; kindly delete
2. Kindly write all the limitations of the current study
3. Write about the clinical implementations
4. Write on clinical recommendations and future research
Conclusion:
Kindly rewrite the conclusion, depending on the preclinical experiment and objectives of the current study, keeping the limitations in mind

---

## Round 0.2 · accepted · Accept

Congratulations to the authors on addressing to the comments of the respected reviewers successfully and increasing the scientific rigour of the paper.

·

Basic reporting

The authors have addressed all the comments, and the manuscript has greatly improved.

Experimental design

The authors have addressed all the comments, and the manuscript has greatly improved.

Validity of the findings

The authors have addressed all the comments, and the manuscript has greatly improved.

Additional comments

The authors have addressed all the comments, and the manuscript has greatly improved.